# Learning collective multicellular dynamics with an interacting mean field neural SDE model

Qi Jiang[1,2], Longquan Li[1,3], Lei Zhang[4], Lin Wan [1,2,3]*

**1** State Key Laboratory of Mathematical Sciences, Academy of Mathematics and Systems Science, Chinese Academy of Sciences, Beijing, China, **2** School of Mathematical Sciences, University of Chinese Academy of Sciences, Beijing, China, **3** School of Advanced Interdisciplinary Sciences, University of Chinese Academy of Sciences, Beijing, China, **4** Department of Control Science and Engineering, Tongji University, Shanghai, China

* lwan@amss.ac.cn

## Abstract

The advent of temporal single-cell RNA sequencing (scRNA-seq) data has enabled in-depth investigation of dynamic processes in heterogeneous multicellular systems. Despite remarkable advancements in computational methods for modeling cellular dynamics, integrating cell-cell interactions (CCIs) into these models remains a major challenge. This is particularly true when dealing with high-dimensional gene expression profiles from large populations of interacting cells, where the intricate interplay between cells can be obscured by data complexity. To address this, we present scIMF, a single-cell deep-generative Interacting Mean Field model that learns collective multicellular dynamics. Leveraging the McKean-Vlasov stochastic differential equation framework, scIMF provides a mathematical foundation for describing interacting multicellular systems, where each cell's evolution depends on the population's empirical distribution. By incorporating a cell-wise attention mechanism, the model efficiently captures nonlocal and asymmetric CCIs, enabling realistic reconstruction of complex intercellular relationships in high-dimensional spaces. Benchmarking across diverse temporal scRNA-seq datasets demonstrates that scIMF outperforms state-of-the-art methods in reconstructing gene expression at unobserved time points and in inferring cellular velocities. Furthermore, scIMF uncovers biologically interpretable, non-reciprocal interaction patterns of cells, providing a principled framework for studying complex, particularly non-equilibrium biological systems.

## Author summary

Interacting particle systems (IPSs) are ubiquitous in nature, from physics to biology, giving rise to complex dynamics at the level of individual constituents and in the system as a whole. While the mathematical theory of IPSs has matured considerably, computational methods lag in learning non-reciprocal interactions

**Data availability statement:** scIMF software is available at https://github.com/QiJiang-QJ/scIMF.

**Funding:** This work was supported by the National Key Research and Development Program of China (No. 2022YFA1004801 to LW) and the Strategic Priority Research Program of the Chinese Academy of Sciences (No. XDB1350203 to LW).The funders had no role in study design, data collection and analysis, decision to publish, or preparation of the manuscript.

**Competing interests:** The authors have declared that no competing interests exist.

from high-dimensional population snapshots, particularly for collective multicellular dynamics in temporal scRNA-seq data. We introduce scIMF, a deep generative model that overcomes this by integrating McKean-Vlasov stochastic differential equations with a cell-wise attention mechanism. Our framework efficiently infers nonlocal, non-reciprocal cell-cell interactions directly from data, outperforming state-of-the-art methods. Crucially, scIMF reveals asymmetric interactions signifying nonequilibrium dynamics in vivo while capturing symmetry in vitro, providing a transformative tool to study collective behaviors in complex dynamical systems.

## Introduction

The emergence of time-series single-cell RNA sequencing (scRNA-seq) data offers a granular view of cellular dynamics, enabling unprecedented resolution of state transitions in heterogeneous populations during development, disease progression, and tissue homeostasis [1]. However, the destructive nature of single-cell profiling inherently disrupts cell-cell correspondences across time points, posing a fundamental challenge for modeling multicellular dynamical systems, where collective behaviors are governed by complex, nonlinear cell-cell interactions (CCIs) [2]. Although temporal scRNA-seq captures snapshots of gene expression profiles over time, integrating CCIs into computational models remains a critical bottleneck, as existing methods often overlook the emergent properties of interacting cell populations.

Numerous efforts have been made to link scRNA-seq snapshots over time [3]. Optimal transport (OT)-based methods like Waddington-OT [4] and generative adversarial frameworks [5] infer probabilistic cell couplings but are restricted to static mappings. Recent advances leverage dynamic optimal transport (DOT) [6] to infer continuous trajectories. For instance, TrajectoryNet integrates DOT with continuous normalized flows [7], while MIOFlow combines stochastic dynamics with a latent manifold structure via a geometric variational autoencoder (VAE) [8]. ScNODE embeds neural ODEs into a VAE with dynamic regularization to handle distribution shifts [9], and PRESCIENT models cell dynamics as stochastic diffusion processes on a potential energy landscape [10]. The physics-informed neural network PI-SDE further extends PRESCIENT by integrating the principle of least action for interpretability and stability [11]. A critical limitation unites these approaches: they operate as single-particle models, where each cell's dynamics depends solely on its own state, ignoring interactions with other coexisting cells. This oversight undermines their ability to predict collective population behaviors, as CCIs are known to drive emergent properties in tissues and tumors [1,2].

Recent studies have begun to address this gap. For example, GraphFP models cell population dynamics using Wasserstein gradient flows with a nonlinear free energy functional, including a quadratic cell-cell interaction term [12]. While interpretable and computationally efficient, GraphFP operates on a discrete state space at the cell type/cluster level, resulting in a coarse-grained view of the system. Similarly, DIISCO infers intercellular interactions at the cell type level using Gaussian processes, incorporating receptor-ligand pairs as a prior [13].

Smart and Zilman propose a Hopfield network-based approach to simulate multiscale cellular dynamics, but their model relies on binarized gene expression and struggles with high-dimensional data [14]. Critically, no existing method efficiently captures high-dimensional, single-cell-resolution dynamics while explicitly modeling cell-cell interactions (CCIs) [15], a key limitation our work aims to resolve.

Here, we propose scIMF, a single-cell deep generative Interacting Mean Field Model for modeling collective multicellular dynamics and quantifying cell-cell interactions from time-series scRNA-seq data. scIMF builds upon the McKean-Vlasov stochastic differential equation (SDE) framework, which provides a mathematical basis for describing the behavior of interacting particle systems [16–19].

Unlike conventional Itô-type SDEs that typically adopt independent particle paradigms, McKean-Vlasov SDE (MV-SDE) models each cell's dynamics as a function of its own state and the distribution of cellular states of the whole population, enabling explicit modeling of intercellular dependencies. By incorporating cell-wise self-attention [20], scIMF captures nonlocal and asymmetric CCI patterns, which are ubiquitous in biological systems far from equilibrium [21,22]. We conduct comprehensive benchmarking of scIMF against state-of-the-art time-series scRNA-seq inference methods (PRESCIENT, MIOFlow, PI-SDE, and scNODE) across three temporal scRNA-seq datasets. Our results demonstrate that scIMF achieves superior performance in (1) reconstructing gene expression at unseen time points with higher accuracy; (2) inferring cellular velocities that better align with observed temporal transitions; and (3) uncovering asymmetric CCI patterns in vivo, a hallmark of out-of-equilibrium biological processes.

## Results

### Overview of scIMF

scIMF is a deep generative framework designed to learn collective cellular dynamics from time-series scRNA-seq data (Fig 1). These data are typically collected at multiple time points from distinct experimental individuals, resulting in population-level snapshots rather than matched cell trajectories. While previous Itô-SDE-based methods (e.g., PRESCIENT) effectively treat each cell as an independent particle, this approach assumes that a cell's fate is determined solely by its internal gene expression state. However, in multicellular systems, cell fate is strictly regulated by the extracellular environment and signals from neighboring cells. To address this limitation, scIMF models multicellular dynamics as interacting diffusion processes governed by a McKean–Vlasov SDE (MV–SDE). This formulation naturally captures system-level behaviors through nonlinear and nonlocal Kolmogorov forward equations [17]. Importantly, this mathematical structure mirrors biological reality: in developing tissues, a cell's trajectory is governed not only by its intracellular gene regulatory network (the intrinsic state) but also by the aggregate signaling milieu. By formally incorporating the empirical distribution into the drift term in Eq (2), scIMF treats the "mean field" as a dynamic proxy for the tissue microenvironment. This enables the model to capture how collective signals exert a guiding force on individual differentiation paths, a feature inherently missing from single-particle formulations.

A central component of scIMF is the integration of Transformer self-attention mechanisms, which effectively model nonlocal dependencies across cells. By treating each cell's gene expression profile as a token, scIMF implements a cell-wise attention module that dynamically infers cell–cell interactions (CCIs) directly from data. The full system is cast within a mean-field game framework [19] and solved efficiently using Neural SDE techniques [23,24], with a Transformer encoder approximating the distribution-dependent drift term of the MV-SDE. Notably, scIMF is jointly driven by the model formulation and the data itself. It requires no prior biological knowledge (e.g., curated ligand–receptor pairs or predefined interaction networks), enabling unbiased inference of complex, possibly asymmetric CCIs in high-dimensional cellular populations.

scIMF offers three principal advantages. (1) Collective dynamics modeling: MV-SDEs provide a mathematical foundation for interacting particle systems, enabling joint inference of temporal dynamics and CCIs. (2) High-dimensional and large-scale data scalability: scIMF is a mesh-free method that scales efficiently to the high dimensionality of gene

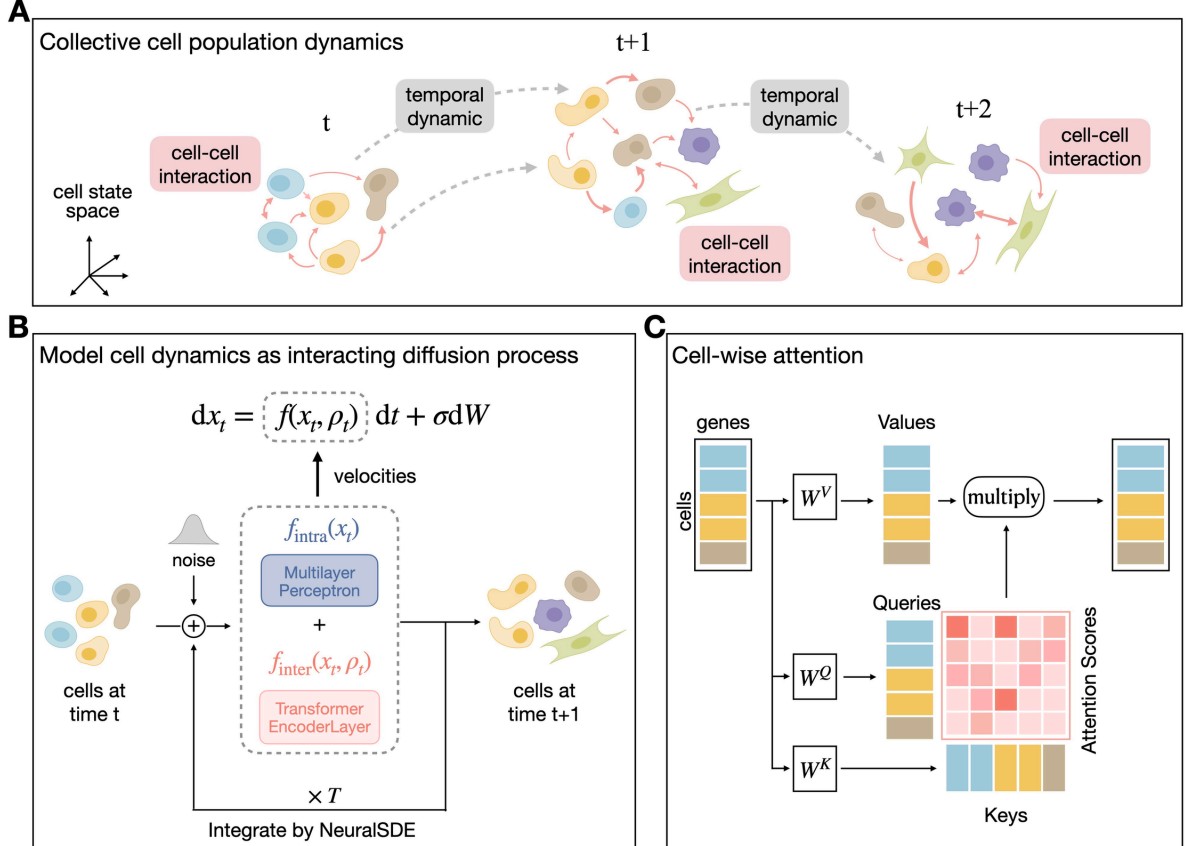

**Fig 1**. **Overview of scIMF.** (A) For time-series data describing the collective cell population dynamics, the destructive nature of scRNA-seq techniques results in a lack of cell-cell correspondence both within time point and across time points. (B) scIMF models complex multicellular dynamics as interacting diffusion processes, described by the MV-SDE, and solves the MV-SDE using the NeuralSDE framework, where the drift coefficient is approximated through two neural networks. (C) scIMF encodes the cell-cell interactions through a cell-wise attention architecture.

expression, and its transformer-based attention mechanism efficiently captures nonlocal dependencies in large cell populations with quadratic complexity. (3) Biological interpretability: The learned attention scores reveal asymmetric interaction patterns, aligning with known features of living systems far from equilibrium [21,22].

To comprehensively evaluate our method, we benchmark scIMF against four state-of-the-art methods, PRESCIENT, MIOFlow, PI-SDE, and scNODE, using three time-series scRNA-seq datasets: (1) zebrafish embryogenesis [25] (ZB data), (2) reprogramming of mouse embryonic fibroblasts to induced pluripotent stem cells [26] (MEF data), and (3) pancreatic $\beta$-cell differentiation [27] (Panc data). The ZB and MEF datasets, collected from in vivo experiments, were previously analyzed in scNODE [9]. The Panc dataset, derived from in vitro experiments, has been used in studies of PRESCIENT [10] and PI-SDE [11].

## scIMF accurately predicts gene expression in vivo

We first evaluate scIMF's predictive performance on two in vivo datasets (ZB [25] and MEF [26]) against four state-of-the-art baselines: PRESCIENT, MIOFlow, PI–SDE, and scNODE. The ZB dataset consists of 38,731 cells sampled across 12 tightly spaced developmental stages (denoted as time points 0, 1, 2, ..., 10, 11), spanning from zygotic genome activation to early somitogenesis. The MEF dataset captures the reprogramming of mouse embryonic fibroblasts (MEFs) into

induced pluripotent stem cells (iPSCs), with 236,285 cells profiled across 39 time points over a 19-day period. Following the preprocessing protocol used in scNODE, we downsampled the MEF dataset to 10% of its original size and performed temporal coarse-graining by grouping the 39 time points into 19 aggregated time points (denoted as $0, 1, 2, \ldots, 17, 18$), where each interval corresponds to a single day over the 19-day span.

To assess each model's ability to predict gene expression at unseen time points, we adopt the held-out evaluation scheme introduced in scNODE [9]. Performance is quantified using the $\ell_1$- and $\ell_2$-Wasserstein distances, $W_1$ and $W_2$, where lower values indicate better agreement between predicted and observed distributions. The held-out schemes are defined as follows: (1) Easy tasks where data at middle time points are removed to evaluate interpolation; (2) Medium tasks, where data at the last few time points are removed to evaluate extrapolation; and (3) Hard tasks, which combines both schemes of easy and medium tasks to assess the model's performance under simultaneous interpolation and extrapolation challenges. The specific held-out time points used in our experiments are: (1) Easy task: ZB - 4,6,8; MEF - 5,10,15; (2) Medium task: ZB - 10,11; MEF - 16,17,18; and (3) Hard task: ZB - 2, 4, 6, 8, 10, 11; MEF - 5, 7, 9, 11, 15, 16, 17, 18.

As shown in Fig 2A-2B, scIMF yields the lowest Wasserstein distances across all benchmark tasks on the ZB dataset. For all three tasks, scIMF outperforms all baselines in both $\ell_1-$Wasserstein ($W_1$) and $\ell_2-$Wasserstein ($W_2$) metrics, with the performance advantage becoming more pronounced in medium and hard tasks involving temporal extrapolation. Notably, in the hard task, scIMF achieves the lowest $W_1$ (12.92) and $W_2$ (15.77), representing 17.2% and 13.5% reductions in Wasserstein distances compared to the second-best model, scNODE ($W_1 = 15.66, W_2 = 18.23$).

On the MEF dataset (Fig 2C-2D), scIMF demonstrates superior performance across three benchmark tasks, with the exception of the hard task evaluated by the $W_2$ metric. Specifically, for the medium task that entails extrapolation across multiple late-time points, scIMF outperforms baselines with $W_1 = 21.14$ and $W_2 = 29.88$, compared to scNODE ($W_1 = 26.51, W_2 = 33.26$) and PI-SDE ($W_1 = 29.05, W_2 = 36.05$). In the hard task, scIMF achieves the lowest $W_1$ score (18.90), though its $W_2$ score (27.73) is slightly higher than scNODE (25.98) and PI-SDE (27.11).

**scIMF recapitulates complex developmental branching and cellular velocities in vivo**

We then visualize the ground-truth gene expressions and model predictions for hard-held-out tasks across datasets using Uniform Manifold Approximation and Projection (UMAP) [28]. As shown in Fig 3, scIMF's predicted gene expression profiles show strong alignment with experimental data, indicating that incorporating cell–cell interactions enables scIMF to capture complex multicellular dynamics. Notably, for ZB data (Fig 3A), characterized by its multibranching developmental trajectories during zebrafish embryogenesis, the scIMF accurately predicts fine-grained cell fates at late stages. By contrast, baseline methods (e.g., scNODE and PI-SDE) fail to resolve the full spectrum of cell fate outcomes, often inferring only a limited number of terminal states.

To further demonstrate scIMF's superiority, we focus on the hard-held-out task using ZB data, which comprises 12 developmental stages and 12 cell types (Fig 4A-4B). We evaluate how cellular velocities inferred by different methods aligned with the expected temporal progression, visualizing predictions at time point $t = 8$ (Fig 4C-4D). Notably, scIMF accurately directs cells toward multiple terminal fates, particularly for neural and eye cells (region within the red dashed circles) and axial cells (region within the red solid circles). In contrast, while PI-SDE and PRESCIENT perform well in the axial cell region, their predictions in the neural and eye regions deviate substantially from the temporal patterns in Fig 4A. Additionally, scNODE and MIOFlow underperform in both regions, with predicted velocities failing to match observed developmental trajectories.

**scIMF reveals biological interpretable and non-reciprocal cell–cell interactions in vivo**

A key feature of scIMF is its use of a cell-wise attention mechanism, which captures intricate intercellular relationships. The attention score at position (*i,j*) quantifies the influence of source cell *j* (key) on target cell *i* (query) at each time point

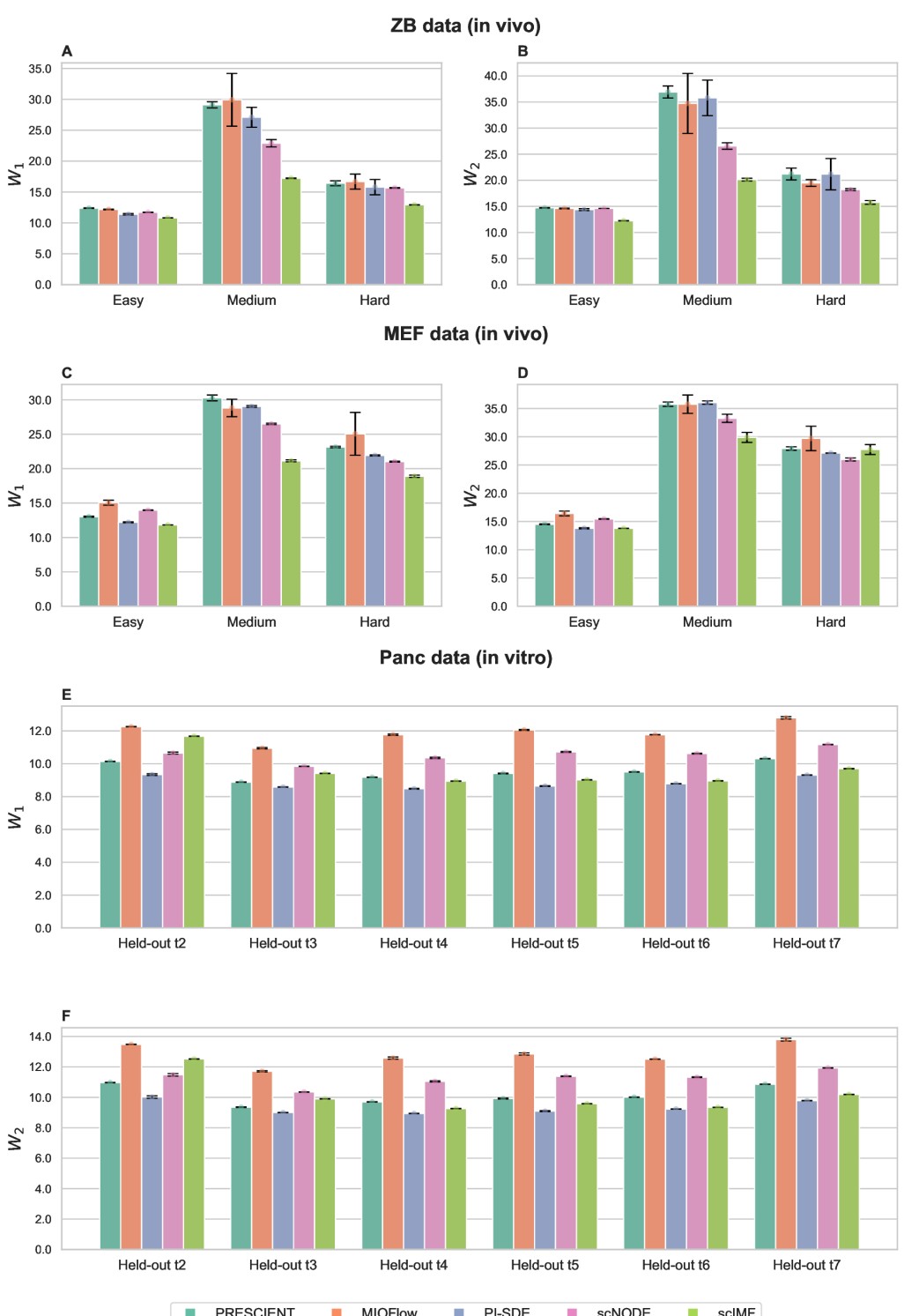

**Fig 2**. **Performance evaluations of scIMF versus baseline methods on held-out tasks.** Quantitative comparisons of scIMF and baseline methods across held-out prediction tasks, measuring the discrepancy between predicted and true gene expression profiles using $\ell_1$- and $\ell_2$-Wasserstein distances ($W_1$ and $W_2$). (A,B) ZB data. (C,D) MEF data. (E,F) Panc data. Each experiment is repeated five times with different random seeds; mean $\pm$ standard deviation is reported for all metrics. Notably, for the ZB and MEF datasets, performance differences in the "Easy" (interpolation) tasks are relatively small across methods, whereas the advantages of explicitly modeling cell–cell interactions (CCIs) in scIMF become most pronounced in the "Medium" tasks, which require future-state (extrapolative) prediction.

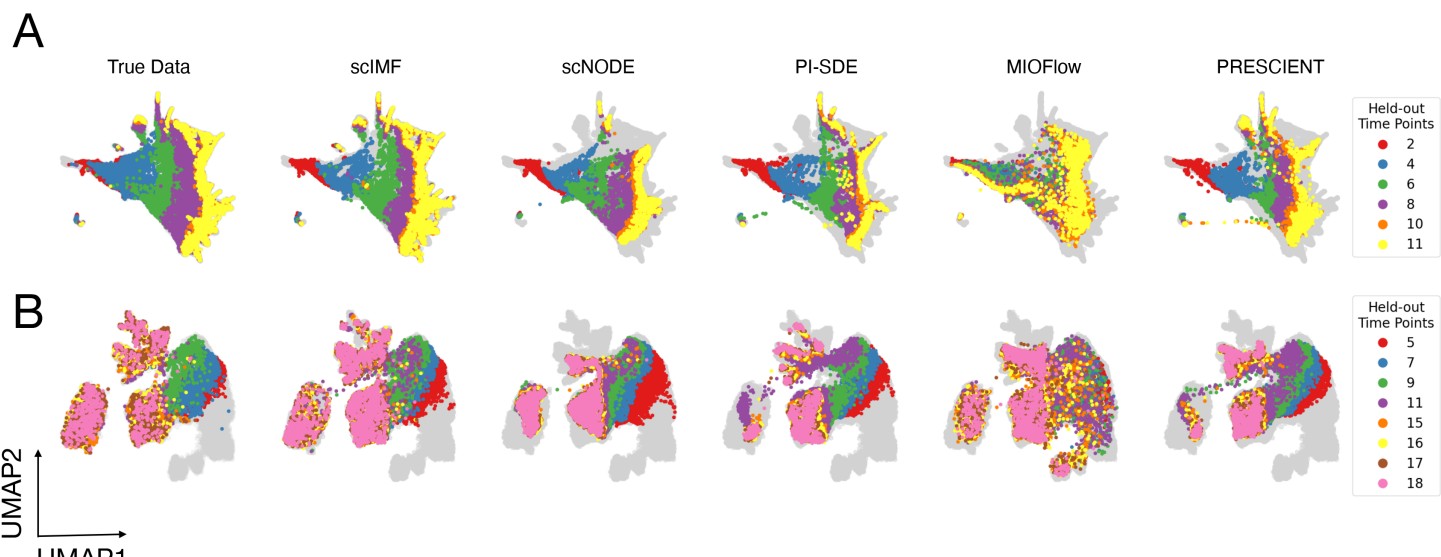

**Fig 3. scIMF reconstructs gene expression in held-out time points with high fidelity.** UMAP visualization of true and predicted gene expressions by scIMF and baseline methods on the held-out unseen time points (color-coded points) in the hard tasks of (A) ZB data and (B) MEF data, with gray points denoting observed training data.

(Fig 1C). When this score approaches zero, it indicates negligible impact of cell $j$ on the dynamics of cell $i$, reflecting minimal interaction or communication between the two cells.

To examine these interactions in detail, we analyze the estimated cell-wise attention matrices across all developmental time points of ZB data (Fig 5A), where cells on both axes are grouped and ordered by annotated cell type. A notable pattern is the progressive emergence of clear cell-type-specific block structures as development proceeds. Each block represents directed influences from one cell type to another, highlighting the dynamic evolution of intercellular communication. Importantly, the attention matrices are highly non-symmetric, enabling scIMF to capture non-reciprocal cell–cell relationships and revealing directionality in communication dynamics. Such asymmetric interaction patterns indicate that the in vivo ZB system operates far from equilibrium or in an out-of-equilibrium regime [21,22].

We then take a close look at cell-wise attention scores over time (Fig 5A). In early development ($t = 0, 1, 2, 3$), intra-cell-type attention scores often show high heterogeneity, characterized by high variance in attention weights within specific cell-type blocks. For example, blastomere subpopulations exhibit diverse interaction patterns, suggesting complex early-stage dynamics where cells of the same annotation may possess varying communicative potentials. By late stages ($t = 7, 8, 9, 10$), intra-cell-type interactions become more homogeneous, appearing as visually uniform blocks, particularly in neural cells; this reflects their transition to functional specialization and synchronized collective behavior. These results align with prior studies demonstrating that cells of the same type may occupy distinct microenvironmental niches, driving context-dependent cell–cell interaction events that orchestrate lineage-specific functional programs [29]. In addition, the inter-cell-type interactions are most active during early zebrafish embryogenesis ($t = 0$ to $t = 5$) and diminish thereafter, indicating reduced cross-lineage communication as development progresses. Collectively, these results reveal a shift from global inter-cell-type communication in early development to cell-type-specific attention patterns, mirroring biological progression from pluripotency to specialization.

To examine interaction patterns at the cell-type level, we compute mean attention scores aggregated by annotated cell types (Fig 5B). Specifically, we construct these directed type-to-type interaction networks by summarizing the cell-wise attention matrices from Fig 5A. In each graph, arrows denote the directional influence from a source cell type to a target

PLOS Computational Biology

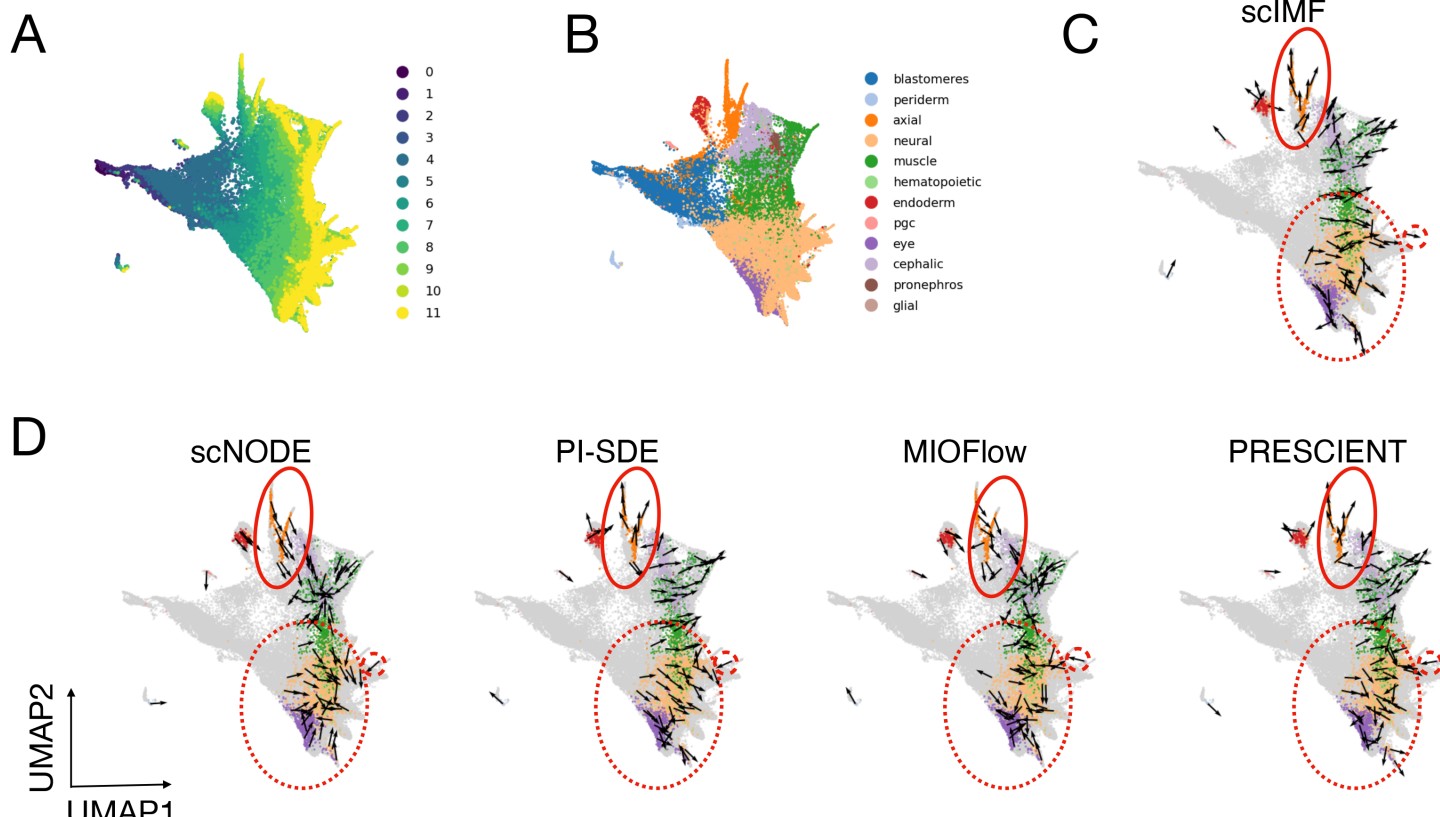

**Fig 4**. **scIMF infers cellular velocities that better align with observed temporal transitions on ZB data.** (A) UMAP of zebrafish embryogenesis cells, colored by developmental time points. (B) UMAP of zebrafish embryogenesis cells, colored by cell types. (C,D) Velocities for a random sample of observed cells at $t = 8$ inferred by scIMF and other baseline methods.

cell type; arrow colors correspond to the source type, and arrow widths are proportional to the magnitude of the mean attention score. During early stages ($t = 0$ to $t = 3$), attention patterns are characterized by widespread interactions among early lineages (blastomeres, periderm, axial), exhibiting strong bidirectional scores that reflect broad regulatory potential and high developmental plasticity. From $t = 4$ onward, new lineages (neural, muscle, hematopoietic, endoderm) emerge, coinciding with lineage-specific interaction structures. For instance, at $t = 4$ and $t = 5$, neural cells primarily act as signal recipients from muscle and endoderm, while endoderm, hematopoietic, and muscle cells show strong outgoing interactions. Concurrently, attention from blastomeres and periderm declines. By $t = 10$ and $t = 11$, cell-type interactions become homogeneous again with bidirectional patterns, aligning with the uniform cell-wise attention matrices at these time points.

To further evaluate whether the attention scores learned by scIMF capture biologically meaningful cell–cell interactions, we performed a post-hoc comparison with CellChat (version 1.6.1) [30] on the ZB dataset. At each developmental time point, we used CellChat's curated zebrafish ligand–receptor database to compute interaction strengths between annotated cell types, and in parallel summarized scIMF's cell-wise attention scores by averaging over all source–target cell pairs to obtain a cell-type–by–cell-type interaction matrix. For each time point, we restricted the comparison to cell-type pairs exceeding minimal thresholds (CellChat score $\geq 0.005$, scIMF attention score $\geq 0.001$). We focused our analysis on time points $t = 3$ through $t = 11$, as earlier stages ($t \leq 2$) contain too few cell types to support reliable correlation analysis, and quantified agreement using Spearman correlation coefficient (SCC).

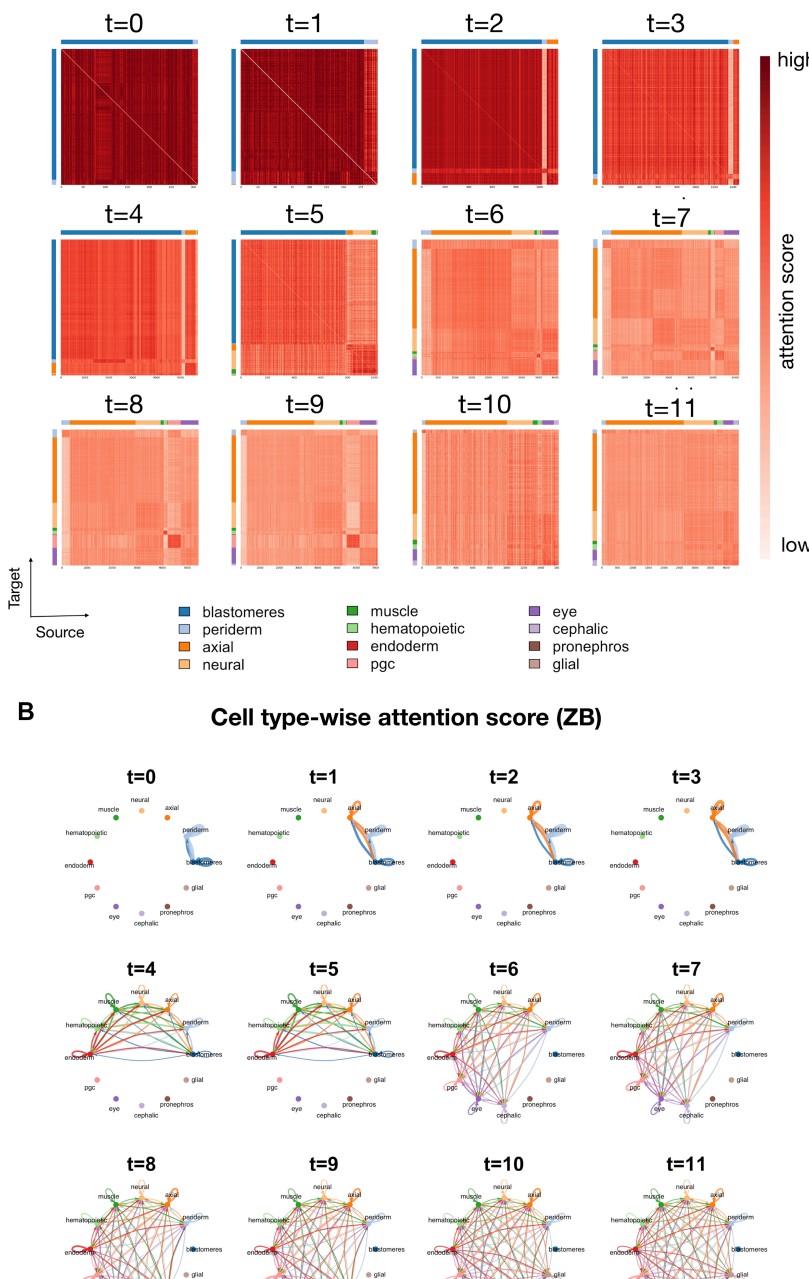

**Fig 5**. **scIMF reveals asymmetric cell–cell interactions in ZB data.** (A) Heatmaps of cell-wise attention score matrices across developmental time points in the ZB dataset. For each matrix, rows correspond to target (queries) and columns to source cells (keys); colors denote the row-normalized attention strength from each source cell to each target cell. Cells along both axes are grouped and ordered by annotated cell type, with colors indicated by the side bars. (B) Cell type–specific directed interaction network obtained by aggregating cell-wise attention scores. Arrows represent directional influence from the source cell type to the target cell type; arrow colors correspond to the source cell type, and arrow widths are proportional to the mean attention magnitude.

The results show heterogeneous patterns across developmental time (S1 Fig). Moderate positive concordance between scIMF and CellChat appears at specific stages (most prominently at $t = 6$ (SCC = 0.55) and $t = 10$ (SCC= 0.41)), where the points form a broadly increasing trend between the two scores. In contrast, early developmental stages ($t = 3$–5) exhibit negligible correlation (SCC $\leq 0.11$), and certain late stages (e.g., $t = 9$, $t = 11$) show weak negative trends (SCC $\approx -0.2$). In these divergent stages, the scatterplots exhibit two characteristic modes. The first is a nearly vertical "upward" pattern (visible in $t = 3$–6), where many cell-type pairs have CellChat scores close to zero but span a wide range of attention scores. This suggests that scIMF identifies putative interactions that may not be captured by the curated ligand–receptor prior, for example when ligand–receptor pairs are missing from the database or sparsely annotated. The second is a shallow, "rightward" pattern (prominent in $t = 9$ and $t = 11$), in which CellChat assigns higher scores but the corresponding attention responses remain relatively flat, likely reflecting ligand–receptor potentials that do not result in active attention signaling in the scIMF model.

Together, these results indicate that scIMF recovers the subset of interactions supported by ligand–receptor evidence at stages of strong concordance, while also highlighting additional communication channels that may be missed or down-weighted by prior-based methods.

### scIMF accurately predicts gene expression and reveals symmetric cell-cell interaction patterns in vitro

To evaluate the applicability of scIMF in a distinct biological setting, we further assess its performance on an in vitro time-series scRNA-seq dataset of $\beta$-cell differentiation (Panc data [27]). The Panc dataset provides a detailed transcriptomic view of stage 5 pancreatic $\beta$-cell differentiation, comprising 51,274 cells assigned to 12 annotated cell types and spanning eight time points from Day 0 to Day 7 (denoted as $0, 1, 2, \ldots, 6, 7$).

We perform held-one-timepoint-out prediction tasks by sequentially withholding each time point beyond the first two and evaluating each model's ability to reconstruct the corresponding gene expression distribution. We compare scIMF with all baseline methods under the same setting (Fig 2E–2F). Overall, PI-SDE attains the highest prediction accuracy, while scIMF performs comparably, consistently ranking second from time point $t = 4$ onward.

To gain a deeper understanding of this result, we visualize the attention scores estimated by scIMF (Fig 6). As shown in Fig 6A, in contrast to the patterns observed in zebrafish embryogenesis, the cell-wise attention matrices across all time points exhibited a relatively symmetric structure, indicating that cell-cell interactions in this in vitro system are less complex and more homogeneous, the cell-type-wise attention scores in Fig 6B reveal highly consistent interaction patterns from $t = 2$ onward. Subtle changes are observed, primarily driven by the gradual emergence of specific cell populations (e.g., *neurog3_mid* and *neurog3_late*). From $t = 4$ to $t = 7$, the global interaction structures showed minimal variation, aligning with the relatively stationary attention matrix patterns in Fig 6A at the same stages.

Taken together, the underlying dynamics of in vitro $\beta$-cell differentiation appear to approach equilibrium or quasi-equilibrium, driven by reciprocal cell-cell interactions [21,22]. In the homogeneous in vitro environment (Panc dataset), where cell signaling is diluted and niche complexity is low. In such settings, the capacity of scIMF to model CCIs yields a smaller marginal gain over baselines like PI-SDE, as evidenced by their comparable performance in Fig 2E–2F.

Conversely, in vivo systems (e.g., zebrafish and MEF datasets) feature complex microenvironments with strong, asymmetric CCIs that drive the system far from equilibrium. Because scIMF captures these population-level influences via an attention-based mean-field term, it demonstrates superior performance in these highly dynamic settings (Fig 2A–2D).

## Discussion

Modeling collective multicellular dynamics remains a central challenge in systems biology. The destructive nature of scRNA-seq techniques precludes the direct observation of cell–cell correspondences, both within a single snapshot and across temporal distinct time points, significantly complicating the reconstruction of dynamics. To address this, we propose scIMF, a deep generative interacting mean-field framework that jointly infers cellular dynamics and quantifies

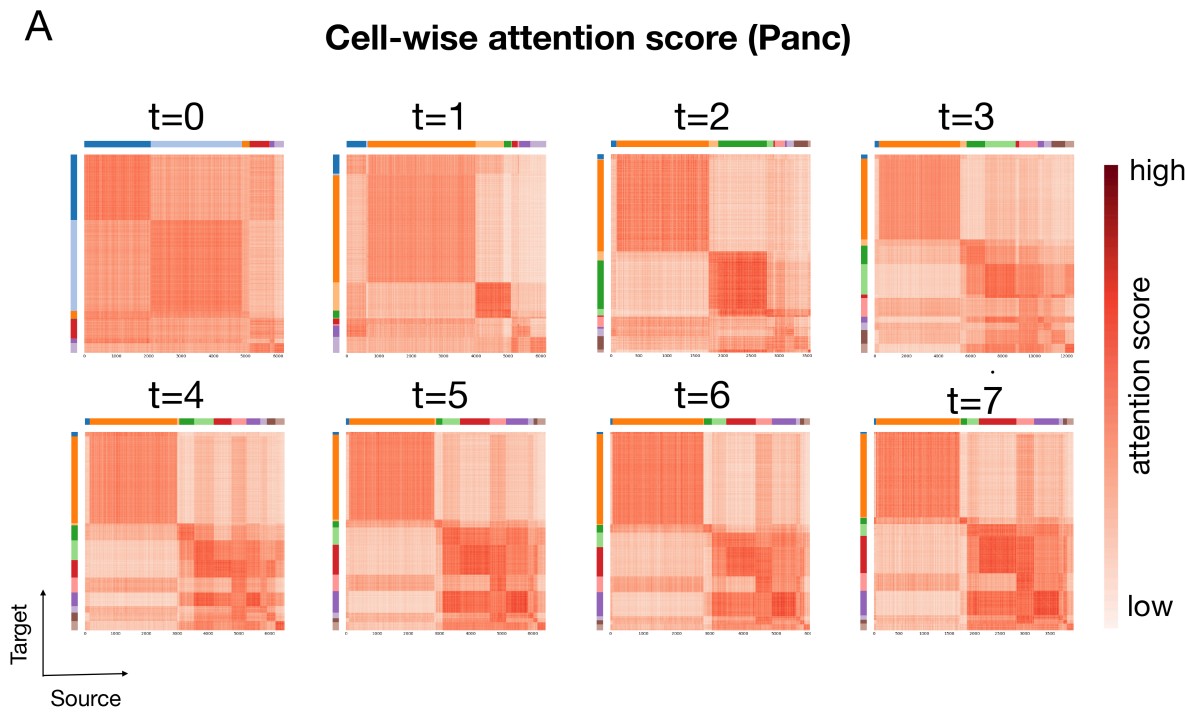

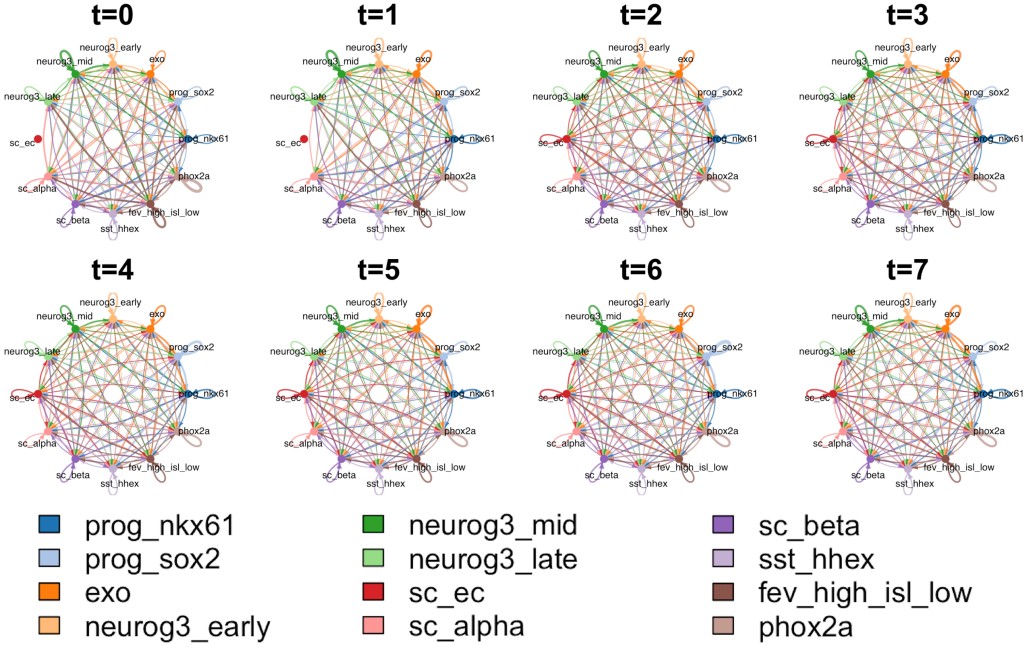

**Fig 6. scIMF reveals symmetric cell-cell interactions in Panc data.** (A) Heatmaps of cell-wise attention score matrices across developmental time points in the Panc dataset. For each matrix, rows correspond to target cells (queries) and columns to source cells (keys); colors denote the row-normalized attention strength from each source cell to each target cell. Cells along both axes are grouped and ordered by annotated cell type, with colors indicated by the side bars. (B) Cell type–specific directed interaction network obtained by aggregating cell-wise attention scores. Arrows represent directional influence from the source cell type to the target cell type; arrow colors correspond to the source cell type, and arrow widths are proportional to the mean attention magnitude.

cell–cell interactions from time-series scRNA-seq data. By implementing McKean–Vlasov SDEs (MV-SDEs) where the drift term depends on the empirical cell-state distribution, scIMF moves beyond single-particle Itô-SDE approaches. Instead, it explicitly models how individual fate decisions emerge from collective behavior—a system-level property that single-particle formulations typically fail to capture.

A key innovation of scIMF is its use of a cell-wise attention mechanism to parameterize these interactions. Unlike conventional neural networks where nonlinearity arises solely from activation functions, our attention-based design captures higher-order nonlinearities in the mean-field drift. By integrating the Neural SDE framework with a transformer encoder to approximate these interacting forces, scIMF efficiently models both the temporal evolution of populations and the structure of intercellular relationships. This integration yields an effective and interpretable tool for dissecting collective dynamics at the individual cell level, offering a comprehensive representation of cellular systems.

Although the primary scIMF model is trained in a prior-free manner, its architecture naturally accommodates biological priors within the intercellular drift term. Conceptually, the interaction dynamics can be decomposed as:

$$\mathbf{f}_{inter} = (1 - \alpha)\mathbf{f}_{global} + \alpha\mathbf{f}_{LR}.$$

Here, the global component $\mathbf{f}_{global}$ corresponds to our current implementation, in which intercellular influences are inferred purely from data by aggregating gene expression across the population. The additional ligand–receptor term, $\mathbf{f}_{LR}$, provides a principled mechanism for incorporating curated biological knowledge. Concretely, given cell-by-ligand and cell-by-receptor matrices, $\mathbf{f}_{LR}$ can be implemented as a cross-attention module mapping ligand profiles of sender cells to receptor profiles of receiver cells. Curated interaction databases are then used to mask or reweight the attention probabilities, ensuring that only biologically plausible sender–receiver pairs contribute to the overall drift. The contribution of this prior-guided branch is modulated by a mixing coefficient $\alpha$; when this coefficient is set to zero, the model recovers the exact formulation of the original, prior-free scIMF.

Nevertheless, several computational and modeling challenges remain that warrant further investigation.

First, the current formulation of scIMF adopts a homogeneous mean-field approximation, where the intercellular drift term $\mathbf{f}_{inter}$ is parameterized by a global attention encoder. Consequently, the same interaction kernel is implicitly applied across all cells and time points. While effective for standard scRNA-seq data, this design does not explicitly leverage modality-specific structures, such as spatial coordinates from spatially resolved transcriptomics or clonal relationships from lineage tracing. A key challenge moving forward is to integrate these complementary data modalities to impose structured constraints on the cell–cell relationship graph. Developing methods that incorporate such prior knowledge, while preserving the flexibility of attention-based architectures, remains an open and promising research direction for enhancing biological fidelity.

Second, our formulation employs a fixed, constant diffusion coefficient $\sigma$ in Eq (2), implying uniform noise levels across the population. Although this choice promotes numerical stability in Neural SDE solvers [31], it potentially oversimplifies the inter-cell/state-dependent variability inherent in transcriptomic dynamics. In future work, we plan to introduce a state-dependent diffusion term $\sigma(\mathbf{x}_t)$ to better capture heterogeneous stochasticity across distinct cell states, alongside the development of robust numerical methods to accommodate this increased complexity.

Third, the present scIMF framework relies on standard optimal transport (OT), which assumes the conservation of total probability mass over time. Consequently, the model does not explicitly account for cell proliferation or apoptosis, which may introduce bias when the underlying biological system undergoes significant expansion or contraction. A principled solution is to extend the framework to unbalanced OT formulations, such as the Wasserstein–Fisher–Rao distance employed in TIGON [32]. We view this as a natural evolution of scIMF and plan to explore this integration in future iterations.

## Materials and methods

### scIMF models multicellular dynamics as interacting diffusion processes

Suppose the time series single-cell samples are collected at $(T+1)$ time points given by

$$\left(t_0, \mathbf{X}^0\right), \left(t_1, \mathbf{X}^1\right), \cdots, \left(t_T, \mathbf{X}^T\right), \tag{1}$$

where $\mathbf{X}^l = \{\mathbf{x}_j^l\}_{j=1}^{N^l} \in \mathbb{R}^{N \times d}$ is a set of cells at a $d$-dimensional space either at the original gene expression space or the low-dimensional space from dimension reduction at time $t_l$ ($0 \leq l \leq T$). At each observed times, cells are assumed to be drawn from a probability distribution in $d$-dimensional gene expression space, denoted by $\{\hat{\rho}_{t_i} \in \mathcal{P}(\mathbb{R}^d) : i = 0, 1, \cdots, T\}$. Here, $\rho_t$ (or equivalently $\rho(\mathbf{x}, t)$) denotes the population-level probability distribution of cellular state $\mathbf{x}$ at time $t$.

To model the collective behavior of the complex multicellular system, scIMF applies an interacting nonlinear diffusion process $\{\mathbf{x}_t \sim \rho_t : t_0 \leq t \leq t_T\}$ described by the MV-SDE:

$$
\begin{aligned}
\mathrm{d}\mathbf{x}_t &= \mathbf{f}(\mathbf{x}_t, \rho_t)\,\mathrm{d}t + \sigma\,\mathrm{d}W_t, \quad t \in [t_0, t_T], \quad x_{t_0} \sim \rho_{t_0}, \\
\rho_t &= \mathrm{Law}(\mathbf{x}_t), \quad \forall t_0 \leq t \leq t_T,
\end{aligned}
\tag{2}
$$

where $\mathbf{f}(\mathbf{x}_t, \rho_t)$ is the interacting mean-field drift term, $\{W_t\}_{t_0 \leq t \leq t_T}$ is a standard Brownian motion, $\sigma \geq 0$ is the strength of diffusion term, $\mathrm{Law}(\mathbf{x}_t)$ represents the probability distribution of the random variable $\mathbf{x}_t$.

Mathematically, the drift term $\mathbf{f}$ for a particular cell is not only related to its own gene expression. Rather, the distribution of cells within the same time point is used as input to account for the influence of cell-cell interactions on cell fate decisions. Furthermore, the nonlinear dependence on $\rho_t$ is nonlocal, as $\mathbf{f}$ is a function not of the value $\rho(\mathbf{x}, t)$ but of the distribution $(\rho(\mathbf{x}, t))_{\mathbf{x} \in \mathbb{R}^d}$ of cells at time point $t$. Note that when the drift coefficient $\mathbf{f}$ is independent of $\rho_t$, Eq (2) reduces to describe a standard (Itô-type) diffusion process. Thus, the MV-SDE generalizes classical SDEs by incorporating distribution-dependent dynamics, allowing for the modeling of cell-cell interactions.

We follow Mishura and Veretennikov [33] and decompose $\mathbf{f}$ into a non-interacting component and an interacting component, with the dependence on $\rho_t$ expressed as an expectation:

$$
\begin{aligned}
\mathbf{f}(\mathbf{x}_t, \rho_t) &= \mathbf{f}_{\mathrm{intra}}(\mathbf{x}_t) + \mathbf{f}_{\mathrm{inter}}(\mathbf{x}_t, \rho_t), \\
&= \mathbf{f}_{\mathrm{intra}}(\mathbf{x}_t) + \int_{\mathbf{y}_t} \varphi\,(\mathbf{x}_t, \mathbf{y}_t)\,\rho_t(\mathrm{d}\mathbf{y}_t).
\end{aligned}
\tag{3}
$$

Here, the first term $\mathbf{f}_{\mathrm{intra}} : \mathbb{R}^d \to \mathbb{R}^d$ on the right hand side (RHS) represents the influence of each cell's individual state on the their dynamics. The second term on the RHS is also called "true McKean-Vlasov case" [33], representing the influence of mean-field interactions. More specifically, $\varphi : \mathbb{R}^d \times \mathbb{R}^d \to \mathbb{R}^d$ quantifies the drift impact of an cell in state $\mathbf{y}_t$ on another cell in state $\mathbf{x}_t$. Then, the total impact on the cell in state $\mathbf{x}_t$, i.e., $\mathbf{f}_{\mathrm{inter}}(\mathbf{x}_t, \rho_t)$, is integrating over the whole state space $\mathbb{R}^d$ weighted by the probability value. While this choice of dynamics is not the most general possible, it is sufficiently broad for many applications of interest [34].

### scIMF encodes the interacting drift term of MV-SDE with Transformer's self-attention

Despite the theoretical richness of the MV-SDE, directly solving its full form poses significant computational challenges due to the high dimensionality of real data and complexity of the interaction terms. Therefore, we use neural networks to parameterize the specific form of the coefficient of MV-SDE (Eq (3)). For $\mathbf{f}_{\mathrm{intra}}$, we can simply utilize a two-layer fully connected neural network as the underlying architecture. However, for $\mathbf{f}_{\mathrm{inter}}(\mathbf{x}_t, \rho_t)$, conventional neural networks, which

process each cell's gene expression independently, fails to consider interactions between cells. Here, we utilize the self-attention mechanism of Transformer, which allows for exploring the quadratic relationships between tokens (e.g., cells in this study). By regarding gene expressions of each cell as tokens, we implement a cell-wise attention mechanism to approximate the $\mathbf{f}_{\text{inter}}(\mathbf{x}_t, \rho_t)$.

More precisely, given a set of cells at time $t$, $\mathbf{X}^t = \{\mathbf{x}_j^t\}_{j=1}^N \in \mathbb{R}^{N \times d}$, we conduct the cell-wise attention mechanism through:

$$
\begin{aligned}
\mathbf{Y}^t &= \text{Attention}\left(\mathbf{Q}^t, \mathbf{K}^t, \mathbf{V}^t\right), \\
&= \mathbf{A}^t \mathbf{V}^t, \\
&= \text{softmax}\left(\frac{\mathbf{Q}^t(\mathbf{K}^t)^\mathsf{T}}{\sqrt{d_k}}\right) \mathbf{V}^t,
\end{aligned}
\tag{4}
$$

where

$$
\mathbf{Q}^t = \mathbf{X}^t \cdot \mathbf{W}^Q, \quad \mathbf{K}^t = \mathbf{X}^t \cdot \mathbf{W}^K, \quad \mathbf{V}^t = \mathbf{X}^t \cdot \mathbf{W}^V.
\tag{5}
$$

Here, the parameter matrices $\mathbf{W}^Q \in \mathbb{R}^{d \times d_k}, \mathbf{W}^K \in \mathbb{R}^{d \times d_k}, \mathbf{W}^V \in \mathbb{R}^{d \times d}$ are learned during training, and $\text{softmax}(\mathbf{z}_i) = \frac{\exp(\mathbf{z}_i)}{\sum_j \exp(\mathbf{z}_j)}$. Notably, $\mathbf{W}^Q$ and $\mathbf{W}^K$ are trained independently without constraints enforcing their equality. As a result, the estimated attention scores can be asymmetric, making them well-suited for capturing the influence of source cells on target cell velocities. This further distinguishes our model from kernel-based methods [13,35], where the interaction matrix restricts to symmetric, implying directionless cell-cell interactions.

To decouple $\mathbf{f}_{\text{inter}}$ from $\mathbf{f}_{\text{non\_inter}}$, we ensure that $\mathbf{f}_{\text{inter}}$ only considers the interactions between cells, excluding self-influence. In the practical implementation, we achieve this by masking the diagonal elements of the attention matrix, ensuring that the attention scores for the diagonal elements (i.e., the attention scores for each cell with respect to itself) are zeroed out.

In addition, interactions between cells can be complex and involve a variety of types (pathways). This can be represented by the multi-head mechanism in Transformer. Specifically, suppose there are $H$ interactions among the multicellular system, the multi-head self-attention is calculated as follows:

$$
\mathbf{Y}^t = \text{Concat}(\mathbf{head}_1^t, \dots, \mathbf{head}_H^t)\mathbf{W}^O,
\tag{6}
$$

where

$$
\begin{aligned}
\mathbf{head}_h^t &= \text{Attention}\left(\mathbf{Q}_h^t, \mathbf{K}_h^t, \mathbf{V}_h^t\right), \\
\mathbf{Q}_h^t = \mathbf{X}^t \cdot \mathbf{W}_h^Q, \quad \mathbf{K}_h^t &= \mathbf{X}^t \cdot \mathbf{W}_h^K, \quad \mathbf{V}_h^t = \mathbf{X}^t \cdot \mathbf{W}_h^V, \\
h &= 1, 2, \cdots, H.
\end{aligned}
\tag{7}
$$

Here, $\mathbf{W}^O \in \mathbb{R}^{Hd \times d}$ is a linear transformation used to integrate the outputs of all attention heads. Each attention head contains its own weights $\{\mathbf{W}_h^Q \in \mathbb{R}^{d \times d_k}, \mathbf{W}_h^K \in \mathbb{R}^{d \times d_k}, \mathbf{W}_h^V \in \mathbb{R}^{d \times d}\}$, which can be learned from data. The outputs of all parallel attention heads are concatenated into a single matrix and transformed by the linear matrix $\mathbf{W}^O$ to give the output matrix $\mathbf{Y}^t$.

The resulting matrix $\mathbf{Y}^t$ is then passed through a subsequent Feed-Forward Network (FFN), which consists of two linear transformations with a nonlinear activation function applied in between. The final outputs approximate $\mathbf{f}_{\text{inter}}(\mathbf{x}_t, \rho_t)$, considering cell-cell interactions.

## scIMF resolves the interacting diffusion process under the mean field game framework

scIMF formulates the learning processes as a mean field game problem [19]. Denote the parametrized neural network for $\mathbf{f}(\mathbf{x}_t, \rho_t)$ as $\mathbf{f}_\theta$, Eq (2) is further solved by optimizing the dynamic OT-like cost:

$$\inf_\theta \mathbb{E}\left\{\int_{t_0}^{t_T} \|\mathbf{f}_\theta(\mathbf{x}_t, \rho_t)\|^2 \mathrm{d}t\right\},$$

$$\text{s.t.} \quad \mathrm{d}\mathbf{x}_t = \mathbf{f}_\theta(\mathbf{x}_t, \rho_t)\,\mathrm{d}t + \sigma \mathrm{d}W_t, \quad t \in [t_0, t_T],$$

$$\rho_t = \mathrm{Law}(\mathbf{x}_t),$$

$$\mathbf{x}_{t_0} \sim \hat{\rho}_{t_0},$$

$$\rho_{t_l} = \hat{\rho}_{t_l}, \; l = 1, 2, \cdots, T. \tag{8}$$

This involves ensuring that the modeled diffusion process aligns well with the observed data distribution while adhering to the principle of least action [11].

The marginal distribution constraints at observed time points can be overly restrictive, making direct optimization challenging. To address this, these constraints are often relaxed by introducing a penalty term into the objective function. Specifically, a Wasserstein loss is added to quantify the discrepancy between the predicted and true distributions, allowing for a more flexible alignment. This relaxed formulation can be written as:

$$\inf_\theta \mathbb{E}\left\{\int_{t_0}^{t_T} \|\mathbf{f}_\theta(\mathbf{x}_t, \rho_t)\|^2 \mathrm{d}t\right\} + \lambda \sum_{l=1}^{T} W_2\left(\hat{\rho}_{t_l}, \rho_{t_l}\right)^2$$

$$\text{s.t.} \quad \mathrm{d}\mathbf{x}_t = \mathbf{f}_\theta(\mathbf{x}_t, \rho_t)\,\mathrm{d}t + \sigma \mathrm{d}W_t, \quad t \in [t_0, t_T],$$

$$\rho_t = \mathrm{Law}(\mathbf{x}_t),$$

$$\mathbf{x}_{t_0} \sim \hat{\rho}_{t_0}. \tag{9}$$

where $\lambda \geq 0$ is a hyperparameter to balance the importance of satisfying the principle of least action and minimizing distributional differences at observed time points, $W_2(\mu, \nu)$ represents the $\ell_2$-Wasserstein distance between distribution $\mu$ and distribution $\nu$.

Finally, we formulate the entire problem as an MV-SDE-constrained optimization problem and solve it efficiently using the Neural SDE framework, where a transformer encoder network approximates the interacting mean-field drift term of the MV-SDE. Optimization of $\mathbf{f}_\theta$ is conducted using the Adam optimizer, with a batch size of 512. For the architecture of $\mathbf{f}_{\text{intra}}$, we utilize a fully connected 2-layer, 256-unit model employing *leakyrelu* as the activation function. As for $\mathbf{f}_{\text{inter}}$, we employ a single-layer transformer encoder with 2 attention heads and a dropout rate of 0.1 for regularization. The diffusion coefficient is prescribed as a constant, set at 0.1. During training, the model is configured with a learning rate of 0.001, and gradient clipping was applied with a maximum norm of 0.1. The Wasserstein distance is computed using the Sinkhorn algorithm [36], with a scaling of 0.7 and a blur of 0.1.

Since self-attention is $\mathcal{O}(N^2)$ in the number of cells $N$, scIMF adopts a mini-batch strategy to scale to large datasets by sampling $B$ cells per iteration ($B \ll N$ in practice). In each iteration, the computational time complexity is dominated by two components: (i) the transformer-based drift term, which is $\mathcal{O}(B^2)$, and (ii) the computation of the Sinkhorn optimal transport loss between the predicted and observed cell distributions, which is $\mathcal{O}(BN)$. Consequently, for fixed batch size $B$ (and a fixed number of Sinkhorn iterations), the overall complexity of scIMF grows approximately linearly with $N$, since attention is confined to fixed-size batches and the only $N$-dependent term arises from the entropically-regularized OT matching. The memory consumption likewise scales linearly with $N$, as evidenced by the approximately linear increase in time and memory usage with the total number of cells used in MEF dataset (S1 Text, S1 Table and S2 Fig).

We set the number of attention heads to $H = 2$ for all three datasets analyzed in this study; notably, the performance of scIMF is robust to this choice (see S2 Text, S1 Table).

## Data preprocessing

All time-series scRNA-seq datasets used in this study have been previously reported and are publicly available. The dataset of zebrafish embryogenesis [25] was downloaded from https://singlecell.broadinstitute.org/single_cell/study/SCP162. The dataset of reprogramming of mouse embryonic fibroblasts to induced pluripotent stem cells [26] was downloaded from https://broadinstitute.github.io/wot/tutorial/. The dataset of pancreatic $\beta$-cell differentiation [27] was downloaded in the NCBI under accession number GSE114412.

All time-series scRNA-seq datasets in this study were processed using the same preprocessing pipeline. For each task, we first selected the top 2,000 highly variable genes (HVGs) based on the training set and restricted all time points to this 2,000-gene feature space. We then performed UMI normalization (library-size scaling to $10^4$) followed by log transformation. Next, we fit a principal component analysis (PCA) model on the training set and projected all time points into a $r$-dimensional latent space using the top principal components. We set $r = 50$ for all three datasets analyzed in this study; notably, the performance of scIMF is robust to the choice of $r$ (see S2 Text and S3 Fig). Because scNODE operates directly in the HVG space, we trained scNODE in the 2,000-dimensional space and subsequently applied the same PCA transformation to its outputs to obtain 50-dimensional representations for fair comparison with all other methods.

## Supporting information

**S1 Text. Computational cost.**
(DOC)

**S2 Text. Robustness of scIMF to the of attention heads ($H$) and the PCA dimension ($r$).**
(DOC)

**S1 Table. Computational cost of scIMF and compared methods on ZB dataset.** Total runtime and Maximum memory usage of scIMF and compared methods on ZB dataset. Note: For GPU-based methods, peak memory refers to peak GPU VRAM; for CPU-only methods (scNODE, MIOFLOW), it refers to peak CPU RSS.
(XLSX)

**S2 Table. Robustness of scIMF to the number of attention heads $H$.** Evaluated $W_1$ and $W_2$ scores of scIMF for different choices of $H$ on the hard task of the ZB dataset; mean $\pm$ standard deviation is reported for all metrics.
(XLSX)

**S1 Fig. Agreement between scIMF and CellChat cell–cell communication scores across developmental time on the ZB dataset.** For each developmental time point $t = 3$–$11$, scatter plots compare the CellChat ligand–receptor–based interaction weights (x-axis) with the scIMF attention scores (y-axis). Only directed interactions between distinct cell-type pairs that pass the applied filters (CellChat score $\geq 0.005$; scIMF attention score $\geq 0.001$) are shown. Each panel corresponds to a single time point and reports the Spearman correlation coefficient (SCC) quantifying the agreement between the two methods. Note that earlier time points ($t \leq 2$) are excluded due to insufficient numbers of cell types and non-trivial interactions for reliable comparison.
(TIFF)

**S2 Fig. Computational cost of scIMF on the MEF dataset.** Total runtime (left) and peak memory usage (right) of scIMF for different data sizes obtained by subsampling 10%, 20%, 50%, and 100% of the 236,285 cells in the MEF dataset.
(TIFF)

**S3 Fig. Robustness of scIMF to the PCA dimension *r*.** Evaluated $W_1$ and $W_2$ scores of scIMF and four comparison methods for different choices of PCA dimension *r* on the hard task of the ZB dataset; mean $\pm$ standard deviation is reported for all metrics.
(TIFF)

## Author contributions

**Conceptualization:** Qi Jiang, Lin Wan.

**Formal analysis:** Qi Jiang, Longquan Li.

**Funding acquisition:** Lin Wan.

**Investigation:** Qi Jiang, Longquan Li, Lei Zhang, Lin Wan.

**Methodology:** Qi Jiang, Lei Zhang, Lin Wan.

**Project administration:** Lin Wan.

**Resources:** Lin Wan.

**Software:** Qi Jiang, Longquan Li, Lei Zhang.

**Supervision:** Lin Wan.

**Validation:** Qi Jiang, Longquan Li.

**Visualization:** Qi Jiang, Longquan Li.

**Writing – original draft:** Qi Jiang, Lin Wan.

**Writing – review & editing:** Qi Jiang, Longquan Li, Lin Wan.

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
