## [Decision Letter · Decision Letter 0]

8 Oct 2025

PCOMPBIOL-D-25-01919

Learning collective multicellular dynamics with an interacting mean field neural SDE model

PLOS Computational Biology

Dear Dr. Wan,

Thank you for submitting your manuscript to PLOS Computational Biology. After careful consideration, we feel that it has merit but does not fully meet PLOS Computational Biology's publication criteria as it currently stands. Therefore, we invite you to submit a revised version of the manuscript that addresses the points raised during the review process.

Please submit your revised manuscript within 30 days Dec 08 2025 11:59PM. If you will need more time than this to complete your revisions, please reply to this message or contact the journal office at ploscompbiol@plos.org. Please include the following items when submitting your revised manuscript:

We look forward to receiving your revised manuscript.

Kind regards,

Yang Lu, Ph.D.

Academic Editor

PLOS Computational Biology

Dimitrios Vavylonis

Section Editor

PLOS Computational Biology

**Journal Requirements:**

At this stage, the following Authors/Authors require contributions: Lei Zhang, Qi Jiang, Longquan Li, and Lin Wan. Please ensure that the full contributions of each author are acknowledged in the "Add/Edit/Remove Authors" section of our submission form.

Potential Copyright Issues:

i) Figures 1a, and 1b. Please confirm whether you drew the images / clip-art within the figure panels by hand. If you did not draw the images, please provide (a) a link to the source of the images or icons and their license / terms of use; or (b) written permission from the copyright holder to publish the images or icons under our CC BY 4.0 license. Alternatively, you may replace the images with open source alternatives. See these open source resources you may use to replace images / clip-art:

**Reviewers' comments:**

Reviewer's Responses to Questions

**Comments to the Authors:**

**Please note that one review is uploaded as an attachment.**

Reviewer #1: This manuscript presents scIMF, a novel deep generative model for learning collective multicellular dynamics from temporal single-cell RNA-seq data. The work is highly timely and addresses a critical gap in the field: the integration of cell-cell interactions (CCIs) into dynamic models of cellular behavior. The authors combine the McKean-Vlasov stochastic differential equation framework with a Transformer-based attention mechanism to capture non-local and asymmetric interactions at single-cell resolution.

Overall, I think this is an interesting work that introduces a valuable and interpretable framework with applicability in gene network dynamics, beyond existing single-cell trajectory inference methods. I have only a few minor questions for the authors to consider:

1, Please add analysis to the time complexity of the proposed algorithm. Is this method applicable to large-scale single-cell RNA-seq datasets? It would be helpful to show the running time.

2, The model infers interactions directly from the data without prior biological knowledge. Could the framework be extended to incorporate known biological priors (e.g., ligand receptor interactions or specific pathway information), and if so, how might that be implemented to potentially improve validity of the algorithm?

3, Line 157, “For example, blastomere subpopulations exhibit diverse interaction patterns, suggesting complex early-stage dynamics. By late stages (t = 7, 8, 9, 10), intra-cell-type interactions become more homogeneous, particularly in neural cells, reflecting their transition to functional specialization.” What’s meaning of this sentence? Is there cell type information in Fig. 5A? Please elaborate the analysis and explain the figure clearly.

Reviewer #2: Summary

The manuscript by Jiang et al. presents scIMF, a novel deep generative model for learning the dynamics of cell populations from time-series single-cell RNA sequencing (scRNA-seq) data. The central contribution of this work is the explicit modeling of cell-cell interactions (CCIs), a factor often ignored by existing methods that treat cells as independent agents. The authors achieve this by framing multicellular dynamics within the McKean-Vlasov Stochastic Differential Equation (MV-SDE) framework, where each cell's trajectory is influenced by the mean field (empirical distribution) of the entire population. To capture specific, non-local, and potentially non-reciprocal interactions, the model innovatively employs a cell-wise attention mechanism. Through comprehensive benchmarking on three distinct datasets, the authors demonstrate that scIMF outperforms current state-of-the-art methods in predicting gene expression at unobserved time points and in inferring cellular velocities. Furthermore, the model's learned attention scores provide biologically interpretable insights, distinguishing between asymmetric, non-equilibrium dynamics in vivo and symmetric, quasi-equilibrium dynamics in vitro.

General Assessment

This is a well-written and technically sophisticated manuscript that addresses a critical and timely challenge in computational systems biology. The integration of mean-field theory from statistical physics with attention mechanisms from deep learning is both novel and powerful. The paper's main strength lies in its principled approach to incorporating CCIs, moving beyond single-particle models to capture the collective behavior that is fundamental to biological processes. The results are compelling, and the demonstrated ability to uncover non-reciprocal interaction patterns is a significant advance with broad implications for studying complex biological systems. The work is of high quality and will undoubtedly be of great interest to the readership of PLOS Computational Biology. I recommend acceptance, contingent on the authors addressing the following points.

Major Comments

Scalability and Computational Complexity: The use of a self-attention mechanism has a computational complexity of O(N^2) with respect to the number of cells (N). While the authors mention the model scales efficiently, this could become a bottleneck for very large datasets (e.g., >100,000 cells per time point). The manuscript would be strengthened by a more detailed discussion of the model's computational performance, including runtime and memory usage as a function of cell number, and a comparison with the baseline models. Is it feasible to apply scIMF to atlas-scale datasets?

Interpretation of the Mean-Field Term: The drift term in the MV-SDE is elegantly split into an intra-cellular component (fintra) and an inter-cellular component (finter). However, the inter-cellular term is modeled entirely by the attention-based transformer encoder. Could the authors elaborate on whether this formulation has any potential limitations? For instance, does it assume all interactions are mediated through a single modality captured by the attention mechanism? A deeper discussion on the biological interpretation and potential constraints of this modeling choice would be beneficial.

Validation of Inferred Interactions: The paper compellingly shows that the inferred attention scores form asymmetric patterns in vivo and symmetric ones in vitro, which aligns with theoretical expectations. This is a powerful, albeit indirect, validation. To further solidify this key claim, could the authors attempt a more direct validation? For example, could the high-attention interactions be cross-referenced with known ligand-receptor pairs or signaling pathways that are expected to be active in those cell types at those developmental stages? While the model is designed to be prior-free, a post-hoc analysis connecting the learned interactions to known biology would significantly enhance the impact and credibility of the findings.

Minor Comments

Figure 2 Clarity: In Figure 2, the performance differences in the "Easy" tasks (interpolation) appear marginal between scIMF and some baselines (e.g., scNODE). The text accurately reflects that the advantage is more pronounced in harder, extrapolative tasks, but it may be worth explicitly noting in the figure caption that the benefits of modeling CCIs are most critical for future-state prediction.

Parameter Sensitivity: The manuscript does not include an analysis of the model's sensitivity to key hyperparameters (e.g., number of attention heads, embedding dimensions). A brief discussion or supplementary figure showing that the model's performance is robust across a reasonable range of hyperparameters would add to the paper's rigor.

Typographical Errors: Please check the manuscript for minor typographical errors. For example, in the held-out task description, "(4) Hard task" should likely be "(3) Hard task".

Reviewer #3: Please see the attachment.

**Have the authors made all data and (if applicable) computational code underlying the findings in their manuscript fully available?**

Reviewer #1: Yes

Reviewer #2: Yes

Reviewer #3: None

PLOS authors have the option to publish the peer review history of their article (what does this mean?). If published, this will include your full peer review and any attached files.

Reviewer #1: No

Reviewer #2: No

Reviewer #3: No

**Figure resubmission:**
---

## [Decision Letter · Decision Letter 1]

12 Jan 2026

Dear Dr. Wan,

We are pleased to inform you that your manuscript 'Learning collective multicellular dynamics with an interacting mean field neural SDE model' has been provisionally accepted for publication in PLOS Computational Biology.

Best regards,

Yang Lu, Ph.D.

Academic Editor

PLOS Computational Biology

Dimitrios Vavylonis

Section Editor

PLOS Computational Biology

Reviewer's Responses to Questions

**Comments to the Authors:**

Reviewer #1: The authors have addressed all my concerns satisfactorily. I have only one minor comment:

Introduction, "...numerous efforts have been made to link scRNA-seq snapshots over time...". I suggest the authors to add a relevant reference here (Chen et al., Reconstructing gene network structure and dynamics from single cell data. Bioinformatics, 41(11), btaf598 (2025)).

Reviewer #2: All my questions have been addressed.

Reviewer #3: All of my concerns have been adequately addressed, and I have no further questions at this time.

**Have the authors made all data and (if applicable) computational code underlying the findings in their manuscript fully available?**

Reviewer #1: Yes

Reviewer #2: Yes

Reviewer #3: Yes

PLOS authors have the option to publish the peer review history of their article (what does this mean?). If published, this will include your full peer review and any attached files.

Reviewer #1: No

Reviewer #2: No

Reviewer #3: No

---

## [Editor Report · Acceptance letter]

PCOMPBIOL-D-25-01919R1

Learning collective multicellular dynamics with an interacting mean field neural SDE model

Dear Dr Wan,

I am pleased to inform you that your manuscript has been formally accepted for publication in PLOS Computational Biology. Your manuscript is now with our production department and you will be notified of the publication date in due course.

With kind regards,

Anita Estes
